# NeuralPCG: Learning Preconditioners for Solving Partial Differential Equations with Graph Neural Networks

## Abstract

Fast and accurate partial differential equation (PDE) solvers empower scientific and engineering research. Classic numerical solvers provide unparalleled accuracy but often require extensive computation time. Machine learning solvers are significantly faster but lack convergence and accuracy guarantees. We present Neural-Network-Preconditioned Conjugate Gradient, or NeuralPCG, a novel linear second-order PDE solver that combines the benefits of classic iterative solvers and machine learning approaches. Our key observation is that both neural-network PDE solvers and classic preconditioners excel at obtaining fast but inexact solutions. NeuralPCG proposes to use neural network models to *precondition* PDE systems in classic iterative solvers. Compared with neural-network PDE solvers, NeuralPCG achieves converging and accurate solutions (e.g., $1e-12$ precision) by construction. Compared with classic solvers, NeuralPCG is faster via data-driven preconditioners. We demonstrate the efficacy and generalizability of NeuralPCG by conducting extensive experiments on various 2D and 3D linear second-order PDEs.[1]

## 1 Introduction

Partial differential equations (PDEs) are fundamental mathematical models with broad applications in science and engineering, for example, the Navier-Stokes equation in fluid dynamics, Poisson's equation in computational geometry, and the Black–Scholes equation in mathematical finance. Despite their powerful modeling ability and wide applications, it is notoriously difficult to find analytical solutions to a general PDE. Therefore, numerical solvers have long been the mainstay of solving PDEs. Classic PDE solvers provide accurate solutions to well-understood PDEs but typically at the cost of long computation time. Speeding up these classic solvers is non-trivial and often requires complex numerical techniques, e.g., multi-grid methods (Briggs et al., 2000), domain decomposition (Smith, 1997), and model reduction (Holmes et al., 2012). Recently, several pioneering works (Li et al., 2018; Sanchez-Gonzalez et al., 2020) introduce machine learning techniques to solving PDEs, particularly in the field of physics simulation. While this line of methods typically outperforms classic solvers in speed by a large margin, it struggles with converging into a highly precise solution (e.g., $1e-12$). A lack of theoretical analysis on convergence and accuracy inhibits neural-network PDE solvers' applications in mechanical engineering, structure analysis, and aerodynamics, where precise PDE solutions have a higher priority than fast yet inexact results.

This work proposes NeuralPCG, a novel and hybrid method that combines the benefits of classic and machine-learning PDE solvers. Our key observation is that neural-network solvers are fast at estimating PDE solutions with a low to moderate accuracy. This property aligns with the *preconditioning* technique in numerical methods, which uses an easy-to-solve approximation of the original PDE to speed up numerical solvers. Based on this intuition, NeuralPCG proposes to learn a neural network that *preconditions* a classic iterative solver. This is in sharp contrast to existing neural-network PDE solvers that replace the numerical solver entirely with a neural network. The backbone of NeuralPCG remains to be a classic solver, empowering it to inherit convergence and

---

[1]Supplementary videos are available on the project webpage: `https://sites.google.com/view/neuralpcg`

accuracy guarantees. Moreover, the learned preconditioner even outperforms classic preconditioners because it adapts to data distributions from target PDE applications.

To demonstrate the efficacy of NeuralPCG, we evaluate it on a set of 2D and 3D representative linear second-order PDEs. We compare its performance with two baselines of very different natures: (1) MeshGraphNet (MGN), a state-of-the-art neural-network PDE solver (Pfaff et al., 2020); (2) the preconditioned conjugate gradient (PCG) algorithm with classic preconditioners; Our experiments show that NeuralPCG has its unique advantage over both baselines: Compared with MGN and similar machine-learning solvers, NeuralPCG generates convergent and accurate solutions by construction and avoids error accumulation. On the other hand, NeuralPCG also outperforms classic PCG solvers in speed because our method learns a preconditioner tailored to the training data distribution, which classic preconditioners typically do not exploit. Finally, NeuralPCG generalizes to physics parameters or meshes with moderate differences ($5\sigma$) from the training data and is robust to outliers by design.

In summary, our work makes the following contributions:

1. We propose NeuralPCG, a novel and efficient solver for linear second-order PDEs that combines the benefits of classic and machine-learning PDE solvers.
2. We present a framework for studying the performance of preconditioners in classic iterative solvers, including a benchmark problem set with carefully designed performance metrics.
3. We conduct extensive experiments to evaluate the efficacy and generalizability of NeuralPCG and demonstrate its advantages over existing methods.

## 2   RELATED WORK

**Numerical PDE solvers**   Classic PDE solvers discretize the continuous PDEs into a numerical system. Popular discretization schemes include finite differences (Strikwerda, 2004), finite elements (Hughes, 2012), and finite volumes (LeVeque et al., 2002). Once the discretized numerical system is formulated, one often needs to solve a linear system. Solving these linear systems accurately and efficiently is the key to a robust PDE solver. Previous work has studied numerical linear algebra extensively (Trefethen & Bau III, 1997; Golub & Van Loan, 2013). These solvers have formed the backbone of many successful, mature numerical PDE software, e.g., ANSYS (Madenci & Guven, 2015), Abaqus (Helwany, 2007), COMSOL (Pryor, 2009), etc. These solvers are crucial for science and engineering, but extensive wall-clock computation time limits their wider deployment. This time bottleneck typically comes from (repeatedly) solving a linear system, and the problem becomes even more severe when the size of the problem scales up. To accelerate linear solvers, prior work has constructed efficient preconditioners for iterative linear solvers via matrix decomposition (Khare & Rajaratnam, 2012), sparsity control (Liu, 1990; Davis & Hager, 1999; Schäfer et al., 2021), and multiscale techniques (Chen et al., 2021).

**Machine learning (ML) methods for solving PDEs**   Researchers have attempted to incorporate machine learning techniques into solving PDEs. A particular line of work focuses on numerical simulation problems, e.g., solving the underlying PDEs for simulating fluid (Kim et al., 2019), cloth (Pfaff et al., 2020), and elastoplastic solids (Sanchez-Gonzalez et al., 2020). Typically, these approaches aim to train a *surrogate* neural network model that replaces the PDE solving process entirely (Brandstetter et al., 2022b;a). Such methods enjoy a major speedup over classic numerical methods thanks to the fast inference time of a trained network model. However, while the network models are capable of producing visually plausible results, they yield no guarantee on the accuracy of the solution. As such, these methods often suffer from error accumulations, inhibiting their wider usage in applications where accuracy is a priority.

**Combining numerical methods with networks**   Given the pros and cons of classic and machine learning methods, researchers have been combining the classical numerical solver with neural networks (Belbute-Peres et al., 2020; Um et al., 2020; Li et al., 2020a;b), with physics-informed neural networks being a particularly notable example (Raissi et al., 2019; Karniadakis et al., 2021). Since neural networks serve as the backbones of these methods, they often have a hard time generating quantitatively accurate solutions (e.g., $1e - 12$ precision) and enforcing hard constraints (Márquez-Neila et al., 2017), both of which are relatively simple with classic numerical solvers. Additionally,

generalization can be an issue when faced with data out of the distribution seen in the training set (Kim et al., 2021). Our method avoids these issues *by construction* because we use a classic solver as the backbone and fine-tune its preconditioner with neural networks. Therefore, convergence and accuracy of the solutions are guaranteed even for unseen data — the performance may degrade but the solving process will not fail. Azulay & Treister (2022) also model preconditioners with neural networks, but their convolutional neural network (CNN) architecture only supports voxel grid discretization and has not been shown to obtain performance gain. Our task and method is perhaps most similar to previous works that uses CNN to learn preconditioners (Sappl et al., 2019; Ackmann et al., 2020). However, we are different from these works in three folds. Becuase CNN architecture only supports grid-based data domains, these methods cannot be directly applied to mesh-based data, whereas we focus on mesh-based problem domains that has several advantages over grid-based data, such as sharp interface handling. Additionally, we propose a novel loss function that exploits the data distribution of field values, but previous methods use condition number as loss energy that only focus on the system matrix. Finally, our proposed method is more generic and works and tested on all second order linear PDEs, whereas previous methods uses hard-coded threshold to satisfy the constraint for a specific problem and is not applicable to other problem settings.

## 3 METHOD

### 3.1 PROBLEM SETUP

In this work, we focus on solving *linear second-order* PDE problems, which cover some of the most common PDEs in scientific and engineering applications, e.g., the heat equation in thermodynamics, the wave equation in acoustics, and Poisson's equation in electrostatics.

**Linear second-order PDEs**   Formally speaking, we write linear second-order PDEs in the following format:

$$\frac{1}{2}\nabla \cdot \mathbf{A}\nabla f(\mathbf{x}) + \mathbf{b} \cdot \nabla f(\mathbf{x}) = c(\mathbf{x}), \quad \forall \mathbf{x} \in \Omega. \tag{1}$$

Here, $\Omega \subset \mathbf{R}^d$ is the problem domain, $f : \Omega \to \mathbb{R}$ is the function to be solved, $\mathbf{A} \in \mathbb{R}^{d \times d}$ and $\mathbf{b} \in \mathbb{R}^d$ are constants, and $c : \Omega \to \mathbb{R}$ is a user-specified function. Without loss of generality, we can safely assume $\mathbf{A}$ to be a symmetric matrix. The definition of $\mathbf{A}$ classifies linear second-order PDEs into *elliptic* (e.g., the Poisson or Laplace equation), *hyperbolic* (e.g., the wave equation), and *parabolic* (e.g., the heat equation) ones. This work studies the model PDEs from these categories.

**Boundary conditions**   We consider a mixture of *Neumann* and *Dirichlet* boundary conditions:

$$\frac{\partial f(\mathbf{x})}{\partial \mathbf{n}} = N(\mathbf{x}), \quad \forall \mathbf{x} \in \partial \Omega_N, \tag{2}$$

$$f(\mathbf{x}) = D(\mathbf{x}), \quad \forall \mathbf{x} \in \partial \Omega_D. \tag{3}$$

Here, $\partial \Omega_N$ and $\partial \Omega_D$ are a partition of the domain's boundary $\partial \Omega$, and $N : \partial \Omega_N \to \mathbb{R}$ and $D : \partial \Omega_D \to \mathbb{R}$ are two user-specified functions. The Neumann boundary condition defines the behavior of the directional gradient of $f$ along the normal direction $\mathbf{n}$ at the boundary, and the Dirichlet boundary condition sets the target value of $f$ at the boundary explicitly.

**Numerical problems after discretization**   The PDE defined above is a continuous problem that one needs to discretize before applying a numerical solver. Discretization approximates gradient operators with numerical computation and converts the continuous PDEs into numerical systems to be solved. Popular discretization schemes include finite difference, finite element, and finite volumes. In this work, we adopt the standard Galerkin method from the finite element theory (Johnson, 2012), resulting in a linear system of equations:

$$\mathbf{K}\mathbf{f} = \mathbf{c}, \tag{4}$$

where $\mathbf{K} \in \mathbb{R}^{n \times n}$, with $n$ being the number of degrees of freedom after discretization, is the *stiffness matrix* of the PDE system after discretization, which is large, sparse, and symmetric-positive-definite (SPD). The vector $\mathbf{c} \in \mathbb{R}^n$ discretizes the function $c$ on the domain and also typically fuses

information from (discretized) boundary conditions. The goal is to solve $\mathbf{f} \in \mathbb{R}^n$, the support of $f$ after discretization at each degree of freedom, from which we can reconstruct the final solution $f$ through numerical interpolation. This linear system $(\mathbf{K}, \mathbf{c})$ becomes the input to our method and other baseline algorithms discussed in this paper.

## 3.2 MOTIVATION

Our strategy is to build a hybrid PDE solver that combines the advantages of both machine learning approaches and classic numerical solvers. Traditional numerical methods for solving $\mathbf{Kf} = \mathbf{c}$ fall into two main categories: direct solvers and iterative solvers. Direct solvers factorize $\mathbf{K}$ into matrices that are easier to solve (e.g., triangular matrices) and are most useful only if the left-hand side matrix $\mathbf{K}$ remains fixed. We are more interested in iterative solvers because they are more suitable for varying $\mathbf{K}$ and $\mathbf{f}$. Iterative solvers, e.g., the conjugate-gradient (CG) method, repeatedly apply matrix-vector products to refine an estimation of the solution until convergence. The runtime of an iterative solver largely depends on the condition number of the stiffness matrix $\mathbf{K}$, motivating the *preconditioning* techniques in numerical methods. At a high level, preconditioning a numerical system means working on a modified system similar to the original problem but faster to solve. It uses the solution of the modified system to bias the iterative solvers towards solving a system with better conditioning, e.g., by clustering the eigenvalues of the stiffness matrix (Solomon, 2015).

Our method is motivated by the observation that the preconditioners described above share remarkable similarities with neural-network PDE solvers (Sanchez-Gonzalez et al., 2020): They both generate inexact solutions with moderate errors in fast computation time. This observation inspires us to propose the Neural-Network-Preconditioned Conjugate Gradient method, or NeuralPCG, which uses CG as the backbone but with a trained neural network as its preconditioner. We see this combination as mutually beneficial for both neural-network methods and classic solvers. On the network side, instead of replacing the whole PDE solver with a neural network as in many previous works (Li et al., 2018; Sanchez-Gonzalez et al., 2020; Pfaff et al., 2020), using a network only to replace the preconditioner ensures convergence, accuracy, and robustness of the solution. It also enables us to generalize the network method to problems that favor precision over speed and to unseen data.

NeuralPCG also benefits CG solvers by proposing a new perspective for designing preconditioners. Designing a high-performance preconditioner is challenging because its two desired properties (fast derivation speed and high similarity to the original system) often conflict. Otherwise, it would imply that the stiffness matrix $\mathbf{K}$ itself was too easy to solve in the first place. The design of classic preconditioners, e.g., Incomplete Cholesky or Symmetric Successive Over-Relaxation (SSOR) (Golub & Van Loan, 2013), is defined on the left-hand-side matrix $\mathbf{K}$ only. We argue that this design decision unnecessarily limits the full power of preconditioners because they overlook the right-hand-side vector $\mathbf{c}$ and its distribution among actual PDE problem instances. In contrast to these classic preconditioners, we propose to learn a neural-network preconditioner from both left-hand-side matrices and right-hand-side vectors in the training data. We expect a higher performance due to such exploitation of data distributions.

## 3.3 THE NEURAL-NETWORK PRECONDITIONER

**Network design**  Given a linear system $\mathbf{Kf} = \mathbf{c}$, our network preconditioner takes as input $(\mathbf{K}, \mathbf{c}$ and outputs a preconditioner $\mathbf{P}$ that CG, a standard iterative solver, can use seamlessly to solve the system. The representative of $\mathbf{K}$ depends on the discretization scheme of the PDEs. For example, a grid discretization leads to a $\mathbf{K}$ matrix conveniently stored as a grayscale image on the grid, and a triangulated PDE results in a $\mathbf{K}$ stored as the node and edge values on a graph. Our work demonstrates our preconditioner on triangle and tetrahedron finite elements and uses GNNs as the network model. However, we believe our core idea is agnostic to discretization schemes.

More concretely, consider a PDE problem on $\Omega$ discretized as a triangle mesh in 2D or a tetrahedron mesh in 3D. We construct a GNN whose nodes and edges are mesh vertices and edges, respectively. We store the input $\mathbf{K}$ as a one-dimensional edge feature: If $\mathbf{K}_{ij}$ is a nonzero entry in $\mathbf{K}$, we add it as an edge feature in the edge from node $i$ to $j$. Similarly, we store vector $\mathbf{f}$ as a one-dimensional node feature on the graph. We then apply the same encoder-message-passing-decoder network architecture as in previous work Pfaff et al. (2020) to predict a one-dimensional feature on each edge, which serves as the output of our network. We leave more details about our network in Appendix.

The last step in our network preconditioner is constructing a valid preconditioner from the network output. Unfortunately, directly assembling the predicted edge feature into a matrix often fails to serve as a valid preconditioner because there is no guarantee of its symmetry or positive definiteness. Therefore, we first construct a lower-triangular matrix $\mathbf{L}(\mathbf{K}, \mathbf{c})$ from the network output, We use $\mathbf{L}\mathbf{L}^\top$ as our preconditioner, and this construction ensures its symmetry and positive definiteness.

**Loss function** Designing many classic preconditioners can be cast as a problem of minimizing their discrepancy to the given linear system over a set of easy-to-compute matrices:

$$\min_{\mathbf{P} \in \mathcal{P}} L(\mathbf{P}, \mathbf{K}) \tag{5}$$

where $\mathbf{K}$ is the stiffness matrix defined above and the system that we want to precondition upon, $\mathcal{P}$ is the feasible set of preconditioners, and $L(\cdot, \cdot)$ is a loss function defined on the difference between the two input matrices. A careful choice of $\mathcal{P}$ ensures the preconditioners remain fast to compute. For example, one can compute the Jacobi preconditioner by choosing $\mathbf{P}$ as the set of all diagonal matrices and $L$ as the difference between the diagonals of the two matrix inputs.

Following the idea described above, it is now tempting to consider the following loss function definition for our neural-network preconditioner:

$$\min_\theta \sum_{(\mathbf{K}_i, \mathbf{f}_i, \mathbf{c}_i)} \|\mathbf{L}_\theta(\mathbf{K}_i, \mathbf{c}_i)\mathbf{L}_\theta^\top(\mathbf{K}_i, \mathbf{c}_i) - \mathbf{K}_i\|_F^2 \tag{6}$$

where $\theta$ is the network parameters to be optimized, $\mathbf{L}(\theta)$ is the lower-triangular matrix from the network's output, and $\|\cdot\|_F^2$ represents the squared Frobenius norm. The index $i$ loops over training data tuples $(\mathbf{K}_i, \mathbf{f}_i, \mathbf{c}_i)$. This definition closely resembles the goal of the famous Incomplete Cholesky preconditioner, especially since $\mathbf{L}$ shares the same sparsity pattern as the lower triangular part of $\mathbf{K}$. However, a closer look at the loss function reveals some potential inefficiencies in its design:

$$L := \sum_i \|\mathbf{L}\mathbf{L}^\top - \mathbf{K}_i\|_F^2 \tag{7}$$

$$= \sum_i \|(\mathbf{L}\mathbf{L}^\top - \mathbf{K}_i)\mathbf{I}\|_F^2 \tag{8}$$

$$= \sum_i \sum_j \|\mathbf{L}\mathbf{L}^\top \mathbf{e}_j - \mathbf{K}_i \mathbf{e}_j\|_2^2 \tag{9}$$

where $\mathbf{e}_j$ stands for the one-hot vector with one at the $j$-th entry and zero elsewhere. This derivation shows that this loss encourages a well-rounded preconditioner with uniformly small errors in all $\mathbf{e}_j$ directions, regardless of the actual data distribution in the training data $(\mathbf{K}_i, \mathbf{f}_i, \mathbf{c}_i)$. Therefore, we consider a new loss function instead:

$$L := \sum_i \|\mathbf{L}\mathbf{L}^\top \mathbf{f}_i - \mathbf{K}_i \mathbf{f}_i\|_2^2 \tag{10}$$

$$= \sum_i \|\mathbf{L}\mathbf{L}^\top \mathbf{f}_i - \mathbf{c}_i\|_2^2. \tag{11}$$

Comparing these two losses, we can see that the new loss replaces $\mathbf{e}_j$ with $\mathbf{f}_i$ from the training data. Therefore, the new loss encourages the preconditioner to ensemble $\mathbf{K}$ not uniformly in all directions but towards frequently seen directions in the training set. Essentially, this new loss trades generalization of the preconditioner with better performance for more frequent data.

**Remark** It is worth revisiting the competing factors in designing high-performance preconditioners and clarifying how our approach handles these factors. Traditionally, designing a preconditioner strikes a balance between fast computation time and similarity with the underlying system, which often conflict with each other. For example, the Jacobi preconditioner is extremely fast to compute. However, it only approximates the diagonal of the system. On the other hand, the Incomplete Cholesky preconditioner well approximates nonzero entries in the system but requires a much longer computation time. Our neural-network preconditioner resolves the conflicts between these factors by inheriting the speed from eural networks and achieves high approximation accuracy by targeting certain data distributions learned from the training data set.

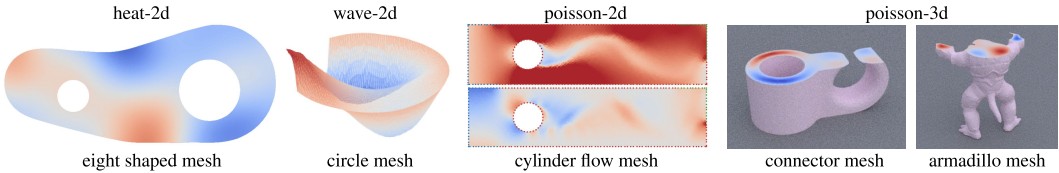

Figure 1: Environment Overview. Left to right: heat-2d, wave-2d, poisson-2d, and poisson-3d.

### 3.4 THE FULL NEURALPCG SOLVER

We are now ready to describe the full NeuralPCG solver during test time. For any $(\mathbf{K}, \mathbf{c})$ input from a PDE instance after discretization and proper boundary handling, we first send the neural network preconditioner $(\mathbf{K}, \mathbf{c})$ to obtain its $\mathbf{LL}^\top$ factorization. Next, we plug the $\mathbf{LL}^\top$ preconditioner into the standard CG solver and run it until the solution reaches a target precision, concluding the full NeuralPCG algorithm. Note that the neural network model only affects the beginning of the CG algorithm by generating the $\mathbf{LL}^\top$ factorization, after which the CG solver no longer needs to access the network. This framework enables a direct and fair comparison between classic CG solvers and NeuralPCG when evaluating their performances: Any performance difference can only come from the difference between preconditioners.

## 4 EXPERIMENTS

Our experiments aim to answer the following questions:

1. How does the proposed method compare with end-to-end network models in speed and accuracy?

2. How does the proposed method compare with classical preconditioners in speed and accuracy?

3. Is the data-dependent loss introduced in the paper effective?

4. Does our approach generalize well to unseen inputs?

We introduce the experiment setup in Sec. 4.1 followed by answering the four questions from Sec. 4.2 to Sec. 4.5.

### 4.1 EXPERIMENT SETUP

**Environments** We provide three environments that study the representative linear second-order PDEs: The **heat** environment studies the heat equation, a parabolic PDE, the **wave** environment studies the wave equation, a hyperbolic PDE, and the **poisson** environment studies the Poisson equation, an elliptic PDE. Each environment is associated with a 2D triangle mesh and/or a 3D tetrahedron mesh for discretization purposes (Fig. 1). More details can be found in the Appendix.

**Baselines** We consider two sets of baselines: neural network and classic numerical methods. For neural network method baseline, we consider the state-of-the art MeshGraphNet (Pfaff et al., 2020) (MGN), which takes the input $(\mathbf{K}, \mathbf{c})$ and outputs a prediction $\mathbf{f}$. For classic solvers, we compare our approach with PCG solvers using two standard preconditioners (Jacobi and IC), which we call **pcg-jacobi** and **pcg-ic**, respectively. We use an all-zero vector as an initial guess for the PCG solver. More details about baselines can be found in the Appendix.

**Our method** The neural network architecture for our preconditioner is based on the MeshGraph-Net. We make modifications of the last layer in order to generate the $\mathbf{LL}^\top$ matrix (see Section 3.3). We then pass in the learned preconditioner into the PCG solver. The only difference between our method and **pcg-jacobi** and **pcg-ic** is the preconditioner. Therefore, any performance difference can be attributed to the preconditioner quality. More details can be found in the Appendix.

**Evaluation Metrics** We consider all baselines and our method as linear solvers: given a pair of $\mathbf{K}$ and $\mathbf{c}$, all methods needs to output $\mathbf{f}$ with the goal of satisfying $\mathbf{Kf} = \mathbf{c}$. The performance of our neural network baseline MGN can be quantified by checking the residual error $\|\mathbf{Kf} - \mathbf{c}\|$. For

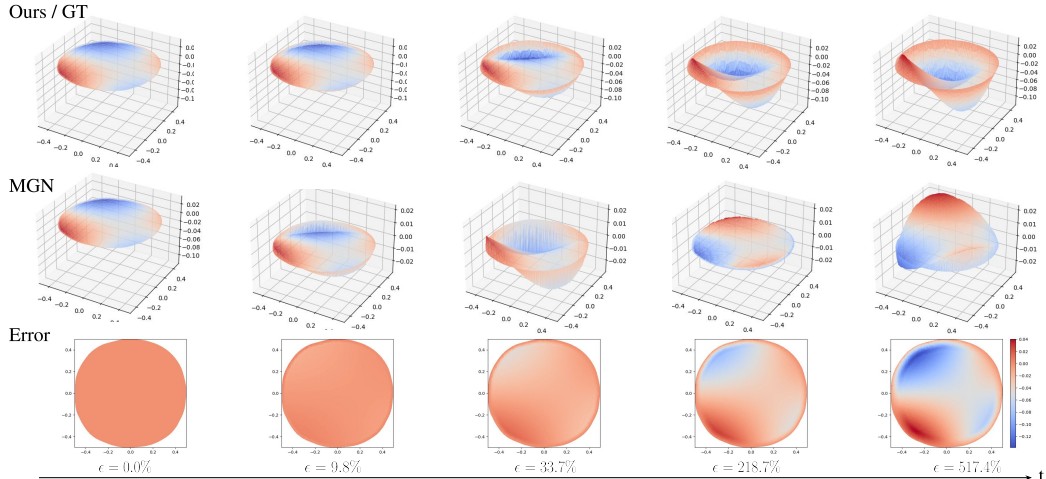

Figure 2: MGN Error accumulation. Field values vs. time step (**wave-2d**): the ground-truth field solved using PCG with 1e-10 convergence threshold (top), the field predicted by MGN (middle), and their difference (bottom), all evaluated at time step 1, 5, 30, 70, 100 (left to right).

PCG baselines and our method, we quantify the performance by comparing the total wall-clock time spent for each preconditioner to reach desired accuracy levels.

To ensure a fair comparison between all methods, we summarize the performance of PCG solvers not in a single number but the following values: (a) the time spent on precomputing the preconditioner for the given $\mathbf{K}$ and $\mathbf{c}$; (b) the iterations and (c) the total time (including the precomputing time) to reach different precision thresholds (e.g., $1e - 12$). We then compare the performance between iterative solvers and end-to-end networks (e.g., MGN) by checking the wall-clock time for the iterative solver to achieve the same residual error as MGN.

### 4.2 COMPARISONS WITH END-TO-END NETWORK MODELS

In the first experiment, we demonstrate one distinct advantage of our approach over end-to-end network methods: We ensure accurate solutions while end-to-end networks accumulate errors over time. To show this quantitatively, we solve the **wave-2d** equation for 100 consecutive time steps using our approach and MGN. Fig. 2 shows that while our approach agrees exactly with the ground truth ($1e - 10$ precision), MGN deviates from the ground truth over time. At time step 100, MGN has an error of $517.4\%$. We can expect MGN to be faster in wall-clock time, as MGN only needs to run network inference once, while we need to run network inference to compute the preconditioner followed by running PCG solvers. Please refer to Sec. A.4.1 in Appendix for detail.

To summarize, MGN is good at estimating solution rapidly while our approach has the flexibility of achieving arbitrary solution precision, just like a standard CG solver. Therefore, network methods are suitable for applications where speed dominates accuracy, while our approach is better for application that requires high precision, e.g., in scientific computing and engineering design.

### 4.3 COMPARISON WITH CLASSIC SOLVERS

Next, we compare our approach with numerical preconditioners, **pcg-jacobi** and **pcg-ic**. Table 8 summarizes the time cost and iteration numbers of PCG solvers using different preconditioners up to convergence threshold 1e-2, 1e-4, 1e-6, 1e-8, 1e-10, and 1e-12.

**pcg-jacobi** conducts the simple diagonal preconditioning so it has little precomputation overhead. However, the quality of the preconditioner is mediocre, and PCG takes many iterations to converge. **pcg-ic** takes fewer iterations to converge. However, its precomputation process is sequential and therefore expensive to compute. By contrast, our approach features an easily parallelizable precomputation stage (like **pcg-jacobi**) and produces a preconditioner with a quality close to **pcg-ic**. Therefore, in terms of total computing time, our approach outperforms both **pcg-jacobi** and **pcg-ic** across a wide range of precision thresholds. We also show additional experimental results in ap-

| Task | Method | Precompute time ↓ (s) | time (iter.) ↓ until 1e-2 | time (iter.) ↓ until 1e-4 | time (iter.) ↓ until 1e-6 | time (iter.) ↓ until 1e-8 | time (iter.) ↓ until 1e-10 | time (iter.)↓ until 1e-12 |
|---|---|---|---|---|---|---|---|---|
| heat-2d | pcg-jacobi | **0.0001** | 0.147 (0) | 1.308 (78) | 2.674 (169) | 3.577 (230) | 4.650 (302) | 5.695 (373) |
| | pcg-ic | 1.119 | 1.321 (**0**) | 1.797 (**32**) | 2.391 (**73**) | 2.781 (**99**) | 3.143 (**124**) | 3.626 (**157**) |
| | Ours | 0.0237 | **0.226** (0) | **0.791** (38) | **1.532** (88) | **2.02** (122) | **2.469** (152) | **3.073** (193) |
| wave-2d | pcg-jacobi | **0.0001** | 0.141 (0) | 0.141 (0) | 0.141 (0) | 0.176 (6) | 0.295 (26) | 0.417 (46) |
| | pcg-ic | 0.7679 | 0.885 (**0**) | 0.885 (**0**) | 0.885 (**0**) | 0.904 (**3**) | 0.953 (**11**) | 1.007 (**20**) |
| | ours | 0.0147 | **0.081** (0) | **0.081** (0) | **0.081** (0) | **0.100** (3) | **0.156** (12) | **0.211** (21) |
| possion-2d | pcg-jacobi | **0.0001** | 0.980 (275) | 1.231 (348) | 1.572 (448) | 1.822 (522) | 2.119 (611) | 2.405 (697) |
| | pcg-ic | 0.7093 | 1.188 (**135**) | 1.309 (**171**) | 1.468 (**219**) | 1.559 **246**) | 1.774 (**311**) | 1.900 (**349**) |
| | ours | 0.0145 | **0.639** (175) | **0.818** (227) | **1.017** (286) | **1.118** (316) | **1.312** (374) | **1.510** (432) |
| possion-3d | pcg-jacobi | **0.0002** | 1.526 (0) | 2.693 (7) | 5.496 (25) | 9.552 (50) | 13.636 (76) | 17.080 (97) |
| | pcg-ic | 3.7371 | 6.441 (**0**) | 6.891 (**3**) | 7.905 (**9**) | 9.436 (**18**) | 11.006 (**27**) | 12.346 (**35**) |
| | ours | 0.4137 | **3.010** (0) | **3.220** (2) | **4.815** (13) | **6.908** (28) | **8.749** (41) | **10.406** (53) |

Table 1: Comparison between preconditioners for PCG. We report precompute time, total time (pcg-icl. precompute time) for each precision level, and the corresponding PCG iterations (in parenthesis). The best value in each category is in bold. ↓: lower the better.

pendix condition number comparison between our method and previous methods in Appendix 7.

## 4.4 ABLATION ON THE LOSS FUNCTION

To highlight the value of our loss function targeting data distributions on the training set, we train the network with the losses in Eqn (6) and in Eqn (10), respectively. Compared to Eqn (6), Eqn (10) exposes the data distribution of **c** to the training process.

| Method | Precompute time ↓ (s) | time (iter.) ↓ until 1e-2 | time (iter.) ↓ until 1e-4 | time (iter.) ↓ until 1e-6 | time (iter.) ↓ until 1e-8 | time (iter.) ↓ until 1e-10 | time (iter.)↓ until 1e-12 |
|---|---|---|---|---|---|---|---|
| Naive loss (Eqn (6)) | 0.0218 | 0.177 (15) | 0.732 (181) | 1.187 (236) | 1.518 (296) | 1.885(296) | 2.284 (361) |
| Our loss (Eqn (10)) | 0.0271 | **0.172** (**14**) | **0.420** (**57**) | **0.597** (**87**) | **0.733** (**110**) | **0.909** (**140**) | **1.056** (**165**) |

Table 2: Wall-clock time and iterations: our method with two different loss functions on **heat-2d**.

Table 2 reports the performances of our loss function Eqn (10) and the naive loss function Eqn (6) on the **heat-2d** environment. We observe that the preconditioner trained with Eqn (10) converges in fewer iterations than Eqn (6). As such, we conclude that enforcing data distribution dependence during training allows us to achieve better in-distribution inference during test time.

## 4.5 GENERALIZATION

**Physics parameters.** First, we consider generalizing the PDEs on their physics parameters, which govern system **K**. We use **poisson-2d** as an example. Our method is trained on a fixed density distribution, and we test the performance of our method on distributions that gradually deviates from the training distribution. Table 3 reports the performance of our approach on these parameters.

Since changing physics parameters does not affect the matrix sparsity, the precomputation time remains largely unchanged across different data distributions (see Table 3 Column 1). Even on the challenging out-of-distribution datasets, our approach still maintains reasonable performance, achieving high precision while using less total time than **pcg-ic** and **pcg-jacobi**.

**Geometry.** We also conduct generalization tests on mesh models, i.e., training network models on one mesh and testing its performance on unseen meshes. We demonstrate this on the **poisson-3d** environment, where we train the network preconditioner on a "connector mesh" (Fig. 1). We then test the trained network model on a new mesh "armadillo" (Fig. 3).

By comparing the bottom and middle rows in Fig. 3, we first see that the end-to-end network method (MGN) struggles to generate accurate solutions when deployed on the unseen mesh (0.5315 error), whereas our approach achieves arbitrary accuracy by construction ($1e - 9$ error threshold here).

We also report the time and iteration cost of our approach and classic preconditioners on **poisson-3d** with armadillo mesh. (Table 4). We notice that our method can generalize to unseen mesh. Even

| Task | Method | Precompute time ↓ (s) | time (iter.) ↓ until 1e-2 | time (iter.) ↓ until 1e-4 | time (iter.) ↓ until 1e-6 | time (iter.) ↓ until 1e-8 | time (iter.) ↓ until 1e-10 | time (iter.) ↓ until 1e-12 |
|---|---|---|---|---|---|---|---|---|
| test | pcg-jacobi | **0.0001** | 0.811 (230) | 0.983 (281) | 1.341 (388) | 1.629 (474) | 1.832 (535) | 2.176 (640) |
|  | pcg-ic | 0.6952 | 1.079 (**100**) | 1.157 (**123**) | 1.275 (**156**) | 1.437 (**203**) | 1.525 (**229**) | 1.648 (**264**) |
|  | ours | 0.0138 | **0.513** (137) | **0.615** (167) | **0.814** (226) | **0.987** (277) | **1.109** (313) | **1.297** (369) |
| test-$\sigma$ | pcg-jacobi | **0.0001** | 0.904 (253) | 1.1 (310) | 1.505 (430) | 1.756 (505) | 2.041 (589) | 2.414 (702) |
|  | pcg-ic | 0.7093 | 1.104 (**110**) | 1.178 (**132**) | 1.338 (**180**) | 1.461 (**217**) | 1.551 (**244**) | 1.715 (**293**) |
|  | ours | 0.0139 | **0.603** (165) | **0.72** (200) | **0.986** (280) | **1.14** (326) | **1.292** (372) | **1.557** (452) |
| test-$3\sigma$ | pcg-jacobi | **0.0001** | 0.949 (262) | 1.139 (317) | 1.566 (441) | 1.828 (519) | 2.123 (606) | 2.485 (714) |
|  | pcg-ic | 0.6992 | 1.114 (**111**) | 1.198 (**137**) | 1.365 (**187**) | 1.484 (**222**) | 1.578 (**251**) | 1.743 (**301**) |
|  | ours | 0.0141 | **0.56** (153) | **0.672** (186) | **0.905** (256) | **1.051** (300) | **1.201** (345) | **1.433** (415) |
| test-$5\sigma$ | pcg-jacobi | **0.0001** | 1.082 (310) | 1.455 (421) | 1.761 (512) | 2.06 (603) | 2.531 (745) | 2.763 (816) |
|  | pcg-ic | 0.6939 | 1.188 (**135**) | 1.311 (**171**) | 1.472 (**219**) | 1.564 (**246**) | 1.783 (**311**) | 1.912 (**349**) |
|  | Ours | 0.0136 | **0.728** (203) | **0.97** (275) | **1.165** (333) | **1.334** (384) | **1.683** (488) | **1.817** (528) |

Table 3: Generalization: physpcg-ics parameters. We test the PCG approaches on testing datasets increasingly deviating from the training data distribution. $\sigma$ stands for standard deviation of training set, test-$\sigma$, test-$3\sigma$, and test-$5\sigma$ means 1, 3, and std. dev away from training distribution, respectively.

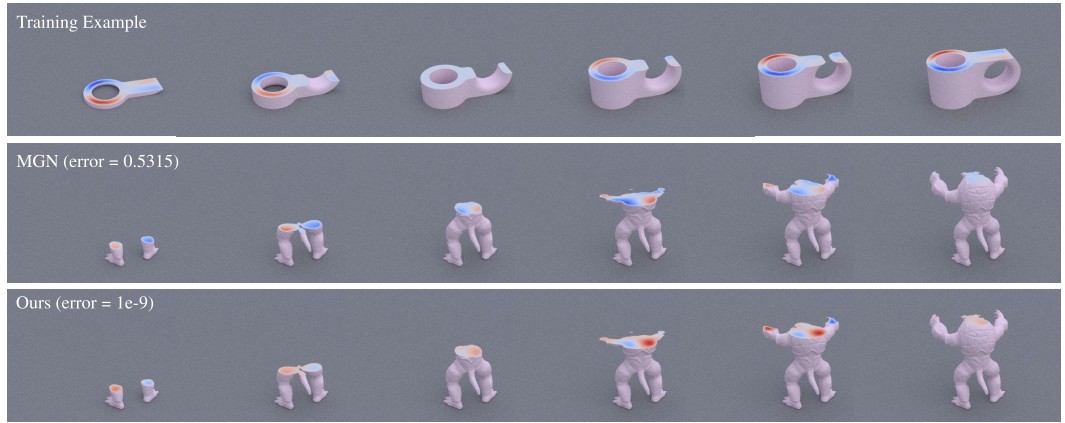

Figure 3: **poisson-3d** on unseen mesh: MGN (middle), our method (bottom), training mesh (top). Left to right: solution fields at different cross-section heights.

in the challenging case of solving PDE on an unseen test mesh, we are still able to converge in less time than **pcg-ic** and **pcg-jacobi** across different precision requirements.

## 5 CONCLUSIONS, LIMITATIONS, AND FUTURE WORK

This work presented NeuralPCG, a hybrid ML-numeric PDE solver. While prior ML PDE solvers often do not have precision guarantees, our approach attains arbitrary precision (up to floating-point error) by construction. Our key observation is that the preconditioner for classic iterative solvers does not require exact precision and is, therefore, an ideal candidate for neural network approximation. NeuralPCG approximates the preconditioner with a graph neural network and embeds this preconditioner into a classic iterative conjugate gradient solver. Compared to end-to-end ML approaches, our approach takes a longer wall-clock time to compute but reduces the error level from

| Method | Precompute time ↓ | Time (iter.) ↓ until 1e-2 | Time (iter.) ↓ until 1e-4 | Time (iter.) ↓ until 1e-6 | Time (iter.) ↓ until 1e-8 | Time (iter.) ↓ until 1e-10 | Time (iter.) ↓ until 1e-12 |
|---|---|---|---|---|---|---|---|
| pcg-jacobi | **0.0002** | 2.314 (6) | 4.634 (22) | 8.395 (48) | 12.002 (72) | 15.381 (95) | 18.332 (115) |
| pcg-ic | 4.9475 | 7.699 (**2**) | 8.362 (**13**) | 9.567 (**23**) | 10.777 (**31**) | 11.891 (**40**) | 12.743 (**50**) |
| ours | 0.4206 | **3.591** (4) | **4.97** (14) | **7.005** (29) | **8.978** (43) | **10.721** (55) | **12.412** (68) |

Table 4: Our approach generalizes to unseen armadillo mesh in the **poisson-3d** environment, and outperforms both **pcg-ic** and **pcg-jacobi** in total wall-clock time.

$1e - 1$ to $1e - 12$. Compared to classic preconditioners, our approach is faster while achieving the same accuracy.

Currently, our approach is limited to linear PDEs. Future work may consider extending to more complex PDEs, such as the elastodynamics equations and the Navier-Stokes equations shown in prior end-to-end ML approaches (Sanchez-Gonzalez et al., 2020). Additionally, we enforce the preconditioner's sparsity to be the lower triangular sparsity of $A$. We consider relaxing this constraint and exploring more effective sparsity control (Schäfer et al., 2021) as an exciting future research direction.

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

## A    APPENDIX

### A.1    ENVIRONMENT DETAILS

Table 5 lists the environment details for experiment section 4.2, section 4.3, section 4.4, section 4.5.

| PDE | Mesh Shape | Number of Nodes | Number of Elements | Physics Parameter | Boundary Condition |
|---|---|---|---|---|---|
| heat-2d | eight shaped mesh | 7454 | 14351 | diffusivity: 10.0 | varying |
| wave-2d | circle shaped mesh | 4852 | 8839 | speed: 0.1 | varying |
| poisson-2d | cynlinder flow | 3167 | 6117 | density: 0.01 | varying |
| poisson-3d | connector mesh | 23300 | 129981 | - | fixed |
| poisson-3d | armadillo mesh | 18181 | 97476 | - | fixed |

Table 5: Environment setup for experiment section 4.2, section 4.3, section 4.4, section 4.5,

Figure 4 shows examples demonstrating varying boundary conditions. For **heat-2d** and **wave-2d**, varying length and position of mesh geometric boundary nodes are selected as Dirichlet boundaries. For **poisson-2d** equation, we use the inviscid-euler fluid equation as a demonstration. All solvers are only responsible for solving the pressure that makes the velocity field incompressible, which is a Poisson equation. The advection and external force steps are then applied to generate the data visualization. For **poisson-2d**, two sets of varying length and position of mesh geometric outer border boundary nodes are selected as influx and Dirichlet boundary. The remaining mesh geometric border nodes, including the remaining outer border and all nodes in the inner border, are obstacle boundary.

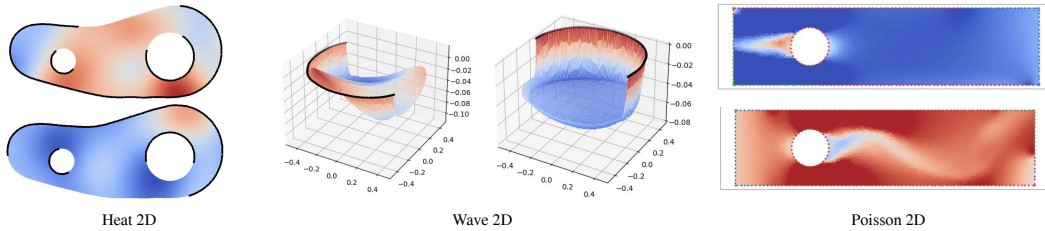

Heat 2D                    Wave 2D                    Poisson 2D

Figure 4: Varying boundary conditions: For **heat-2d** and **wave-2d**, the black vertices represent the Dirichlet boundary. The **poisson-2d** example shows the red vertices are the obstacle boundary, the blue vertices are the influx boundary, and green vertices are the Dirichlet boundary.

For experiment 4.5 across different physics parameters we consider the same mesh domain used in other **poisson-2d** experiments with mesh size of 3167 and element size of 6117. We train on a set of training set with density distribution from [0.001, 0.005], and our test environment *test*-$1\sigma$ is of density 0.006. Test environment *test*-$3\sigma$ is of density 0.008. Test environment *test*-$5\sigma$ is of density 0.01.

### A.2    IMPLEMENTATION DETAILS AND HYPERPARAMETER

In this Section, we describe the implementation details of each method.

#### A.2.1    BASELINES

**MeshGraphNet** To ensure a fair comparison, we re-implement MeshGraphNet in PyTorch, the framework we used for our method, based on their paper and released code. We have also manually tuned its hyperparameters on our dataset to ensure we see reasonably good performance from Mesh-GraphNet. We follow the Encoder-Processor-Decoder. Encoder, Processor and Decoder modules are all Multi-Layer Perceptrons (MLP) with ReLU activation function. Specifically, a 2 layer MLP encoder and a 2 layer MLP decoder are used for the **heat-2d**, **wave-2d**, and **poisson-2d** equation. The Processor Module is composed of 5 iterations of message passing layer. For the **poisson-3d** equation, we use a 2 layer encoder, a 2 layer decoder, and 3 iterations of message passing for the processor module. All MLPs used are of hidden dimension size of 16.

**Classical Methods**

PCG-IC is implemented using pytorch and is fully vectorized for optimized computation. This implementation follows Golub & Van Loan (2013).

PCG-Jacobi is implemented using pytorch and fully vectorized. The Jacobi or diagonal preconditioner is described in Axelsson (1996).

### A.2.2 OUR METHOD

**Technical Details and Justification** We follow the diagonal decomposition $\mathbf{LDL}^\top$ as a way of lower triangular decomposition for the original system $\mathbf{K}$. It is easy to see that this diagonal decomposition is equivalent to lower triangular decomposition.

$$\mathbf{K} = \mathbf{L}_\theta \mathbf{L}_\theta^\top \tag{12}$$
$$= \mathbf{L}_{\theta'} \sqrt{\mathbf{D}} \sqrt{\mathbf{D}} \mathbf{L}_{\theta'} \tag{13}$$
$$= \mathbf{L}_{\theta'} \mathbf{D} \mathbf{L}_{\theta'} \tag{14}$$

The diagonal decomposition $\mathbf{LDL}^\top$ has several advantages, similar to lower triangular decomposition $\mathbf{LL}^\top$, it is easy to invert, and guarantees symmetry. Additionally, we enforce the diagonal element $\mathbf{D}$ to be the diagonal elements of original system $\mathbf{K}$. This way, we enforce the value and gradient range for the lower triangular matrix $\mathbf{L}_{\theta'}$ to ensure positive definiteness of the learned decomposition.

**Implementation Details** We follow the same Encoder-Processor-Decoder architecture implementation for MeshGraphNet. A 2 layer MLP encoder and a 2 layer MLP decoder are used for the **heat-2d**, **poisson-2d**, and **poisson-3d** equation. The Processor Module is composed of 5 iterations of message passing layer. For the Poisson 3D equation, we use a 2 layer encoder, a 2 layer Decoder, and 3 iterations of message passing for the processor module. All MLPs used are of hidden dimension size of 16.

## A.3 TRAINING SPECIFICATION

We use a workstation equipped with 64-core AMD 3995W CPU and a NVIDIA RTX A6000 GPU for all our experiments.

We adopt end-to-end training scheme with the loss functions described in section 3. We train our method for the **heat-2d** equation environment for 6 hours. The **wave-2d** equation is trained for 6 hours. The **poisson-2d** equation is trained for 6 hours. The **poisson-3d** Equation is trained for 8 hours.

We train the NN baselinebaseline for 3 days, 72 hours, for experiments wave-2d, poisson-2d, and heat-2d.

## A.4 ADDITIONAL EXPERIMENTAL RESULTS

### A.4.1 TIME COMPARISON WITH END-TO-END NEURAL NETWORK

We first report in Table 6 the time cost of network methods and that of our method achieving the same residual error. Specifically, for each environment, we first run MGN on the test set. We then record the wall-clock time and the residual error from each pair of $(\mathbf{K}, \mathbf{c})$. Next, we use the residual error as the convergence threshold in NeuralPCG and record the wall-clock time cost. It is unsurprising from the table to see that MGN is substantially faster than our approach: For a given pair of $\mathbf{K}$ and $\mathbf{c}$, MGN only need to run network inference once, while we need to run network inference to compute the preconditioner followed by running PCG solvers. We notice that our method usually converge to the MGN residual error in one iteration.

### A.4.2 CONDITION NUMER

We show the condition number comparison in Table 7.

| env. | heat-2d | wave-2d | poisson-2d | poisson-3d |
|---|---|---|---|---|
| time (MGN) (s) | 0.0213 | 0.0171 | 0.0135 | 0.1154 |
| time (ours) (s) | 0.0950 | 0.0925 | 0.0527 | 1.7762 |

Table 6: The wall-clock time of MGN and our method stopped at the same error produced by MGN.

| Method | Wave-2d | Poisson-2d | Heat-2d | Poisson-3d |
|---|---|---|---|---|
| P (original system) | 540272.25 | 43658.16 | 181.56 | 1008.79 |
| **pcg-jacobi** | 96.35 | 18712.16 | 165.71 | 225.86 |
| **pcg-ic** | 23.22 | 5662.48 | 42.83 | 43.23 |
| Ours | 23.89 | 8384.31 | 64.23 | 136.86 |

Table 7: Condition number comparison between various methods

### A.4.3 ADDITIONAL COMPARISON WITH CLASSICAL PRECONDITIONERS

Here we show additional experimental results on the four equations, **heat-2d**, **wave-2d**, **poisson-2d**, and **poisson-3d**, under other physical parameters or other meshes.

| Task
mesh & parameter | Method | Precompute time
(s) | time (iter.)
until 1e-2 | time (iter.)
until 1e-4 | time (iter.)
until 1e-6 | time (iter.)
until 1e-8 | time (iter.)
until 1e-10 | time (iter.)
until 1e-12 |
|---|---|---|---|---|---|---|---|---|
| heat-2D
circle mesh
diffusivity=10.0 | Jacobi
IC
Ours | **0.0002**
1.0105
0.0271 | 0.240 (27)
1.156 (**12**)
**0.172** (14) | 0.766 (109)
1.346 (**44**)
**0.420** (57) | 1.131 (167)
1.495 (**69**)
**0.597** (87) | 1.406 (212)
1.611 (**88**)
**0.733** (110) | 1.784 (273)
1.751 (**112**)
**0.909** (140) | 2.080 (321)
1.873 (**132**)
**1.056** (165) |
| heat-2D
eight shaped mesh
diffusivity=1.5 | Jacob
IC
Ours | 0.0001
1.5453
0.0251 | 0.669 (32)
1.931 (**13**)
**0.49** (17) | 1.755 (132)
2.369 (**44**)
**1.284** (71) | 2.698 (202)
2.757 (**71**)
**1.856** (110) | 3.471 (257)
3.063 (**93**)
**2.3** (140) | 4.317 (333)
3.346 (**112**)
**2.831** (177) | 5.174 (398)
3.696 (**137**)
**3.377** (214) |
| wave-2D
circle mesh
speed=0.3 | Jacob
IC
Ours | 0.0128
0.7088
0.0171 | 0.072 (0)
0.781 (**0**)
**0.091** (0) | 0.072 (0)
0.781 (**0**)
**0.091** (0) | 0.072 (0)
0.781 (**0**)
**0.091** (0) | 0.105 (6)
0.794 (**2**)
**0.104** (2) | 0.214 (24)
0.843 (**10**)
**0.156** (11) | 0.343 (46)
0.899 (**20**)
**0.214** (21) |
| poisson-2D
circle mesh
density=0.01 | Jacob
IC
Ours | 0.0001
0.9052
0.0143 | 1.339 (219)
1.505 (**88**)
**0.85** (125) | 1.643 (271)
1.632 (**110**)
**1.01** (152) | 2.211 (369)
1.819 (**141**)
**1.316** (202) | 2.670 (449)
2.041 (**178**)
**1.57** (243) | 3.071 (518)
2.172 (**200**)
**1.776** (277) | 3.653 (620)
2.403 (**238**)
**2.076** (325) |
| poisson-2D
cynlinder flow
density=0.0005 | Jacob
IC
Ours | 0.0001
0.6905
0.0145 | 0.980 (275)
1.140 (**115**)
**0.639** (175) | 1.231 (f348)
1.213 (**135**)
**0.818** (227) | 1.572 (448)
1.385 (**183**)
**1.017** (286) | 1.822 (522)
1.504 (**215**)
**1.118** (316) | 2.119 (611)
1.614 (**246**)
**1.312** (374) | 2.405 (697)
1.746 (**282**)
**1.51** (432) |
| possion-3D
armadillo mesh | Jacobi
IC
Ours | **0.0001**
3.3779
0.1142 | 1.749 (8)
5.175 (**2**)
**2.273** (5) | 4.365 (38)
6.038 (**13**)
**3.726** (23) | 6.642 (63)
6.869 (**23**)
**4.919** (37) | 8.995 (89)
7.603 (**31**)
**6.194** (52) | 11.363 (115)
8.336 (**40**)
**7.325** (65) | 13.832 (143)
9.237 (**50**)
**8.677** (81) |

Table 8: Additional experimental results comparing with classical preconditioners.

### A.4.4 ADDITIONAL COMPARISON WITH LEARNING-BASED PRECONDITIONER

We show our proposed method compare with a learning based preconditioner Sappl et al. (2019) in Table 9. Comparing to previous learning-based preconditioner, NN baseline, our method shows stable convergence across various settings using different second order linear PDEs. NN baseline (Sappl et al., 2019) is not orignally designed to be a generic method to work in different problem domains. For example, the hard-coded diagonal clipping threshold of $\epsilon = 1e-3$ is carefully picked by the authors for the urban water problem and there is no general principles to picking such threshold for other problem settings. We can see that NN baseline Sappl et al. (2019) works on par with our method on the wave-2d setting, but it does not work in many other settings, heat-2d, poisson-2d. For Poisson-2d setting the resulting convergence iteration is even worse than the **pcg-jacobi** method. Additionally, the method Using condition number as the loss energy is not computationally efficient. computing condition number for a system requires full eigen decomposition, which is $O(n^3)$. Computation scales cubically with problem size. Our poisson-3d setting uses mesh of size 23,300 nodes. We train the NN baselinemethod for three days, 72 hours, and it only make through less than one percent of the training set, using the same hardware setup. Empirically, we found that using condition number as loss energy is computationally infeasible with problems more than 10,000 nodes.

| Task | Method | Precompute time ↓ (s) | time (iter.) ↓ until 1e-2 | time (iter.) ↓ until 1e-6 | time (iter.) ↓ until 1e-10 |
|---|---|---|---|---|---|
| heat-2d | NN baseline | 0.0236 | 0.226 (0) | 1.973 (116) | 3.223 (200) |
| | Ours | 0.0237 | **0.226** (0) | **1.532** (88) | **2.469** (152) |
| wave-2d | NN baseline | 0.0165 | 0.087 (0) | 0.087 (0) | 0.165 (12) |
| | ours | 0.0147 | **0.081** (0) | **0.081** (0) | **0.156** (12) |
| possion-2d | NN baseline | 0.0129 | 1.014 (291) | 1.602 (471) | 2.221 (662) |
| | ours | 0.0145 | **0.639** (175) | **1.017** (286) | **1.312** (374) |

Table 9: Comparison between preconditioners for PCG. We report precompute time, total time (pcg-icl. precompute time) for each precision level, and the corresponding PCG iterations (in parenthesis). The best value in each category is in bold. ↓: lower the better.

