# OpenReview forum: "NeuralPCG: Learning Preconditioner for Solving Partial Differential Equations with Graph Neural Network"
_ICLR.cc/2023/Conference — Submitted to ICLR 2023_

### Official Review · Reviewer_G2sH · 2022-10-23

**Confidence:** 3
**Correctness:** 3
**Technical Novelty And Significance:** 1
**Empirical Novelty And Significance:** 1
**Recommendation:** 3

**Clarity, Quality, Novelty And Reproducibility:**

Clarity: seems fine.


Quality: not high.


Novelty: very limited.

Reproducibility: Perhaps extremely difficult. The source code is not provided.

**Strength And Weaknesses:**

Strength:

The paper is easy to read and it is straightforward to catch the basic idea of the paper.

Weaknesses:

1. Learning preconditioners using neural networks for improving solving systems of linear equations has been widely studied. However,
*the entire existing works are completely ignored by the authors*. I actually doubt if this paper has any novelty over existing works. By typing "Learning preconditioner" on the google search engine, we immediately get many related papers, e.g., the first two papers are: https://arxiv.org/abs/1906.06925 and https://arxiv.org/abs/2010.02866. Indeed, the study in the first paper is much more solid than the paper under review. Look at figure 5 of the the paper https://arxiv.org/abs/1906.06925, you will see the existing algebraic multi-grid (AMG) preconditioner is much better than the learned ones, Jacobi. and etc. In the paper under review, the authors never contrast the learned preconditioner with the existing state-of-the-art.


2. The authors claimed that "Classic numerical solvers provide unparalleled accuracy but often require extensive computation time.  Machine learning solvers are significantly faster but lack convergence and accuracy guarantees. We present Neural-Network- Preconditioned Conjugate Gradient, or NeuralPCG, a novel linear second-order PDE solver that combines the benefits of classic iterative solvers and machine learning approaches. Our key observation is that both neural-network PDE solvers and classic preconditioners excel at obtaining fast but inexact solutions. NeuralPCG proposes to use neural network models to precondition PDE systems in classic iterative solvers." --- I wonder if there is any convergence and accuracy guarantee of PDE solvers using the learned preconditioner? If not, then the bottleneck of existing learning-based PDE solvers is also a bottleneck of the proposed approach.'


3. The authors stated that "While this line of methods typically outperforms classical solvers in speed by a large margin, it struggles with converging into a highly precise solution (e.g., 1e − 12). A lack of theoretical analysis on  convergence and accuracy inhibits neural-network PDE solvers’ applications in mechanical engineering, structure  analysis, and aerodynamics, where precise PDE solutions have a higher priority than fast yet inexact results." --- My concern is whether we really need 1e-12 accuracy in real applications. Also, if we need such an accurate solution, do learning-based PDE solvers really faster than classical PDE solvers?


4. The authors stated that "NeuralPCG generalizes to physics parameters or meshes with moderate differences from the training data and is robust to outliers by design." --- Does the learned preconditioner generalizes to PDEs of different types and problems at different scales? AMG can be applied to problems of different sizes and even different types. I would like the authors to make some comparisons of their NeuralPCG with AMG and other state-of-the-art preconditioners.


5. Does NeuralPCG works well when to the linear PDE when A is x-dependent?


6. I wonder if NeuralPCG works for solving high-frequency wave equations?


7. The authors may compare the method with some fast methods for solving PDEs, e.g., fast Fourier transformer and fast multipole method. Rather than consider the method that is applicable to any type of PDEs, which is a highly unfair comparison as the learned
preconditioner is problem-dependent.


8. Does the learned preconditioner reduce the condition number? I want to see some numerical evidence of reducing
condition number of the underlying problems.


9. Does over-smoothing of graph neural network affect the performance of the learned preconditioner? The computational cost of
graph neural networks can be non-negligible when the system of linear equations is massive.


10. In equation (6), the learned matrix L can be dense, significantly reducing the scalability of the PDE solver. This can be
a significant bottleneck of NeuralPCG.


11. I think the authors should report the computational time, including the time for learning the preconditioner. In particular, for large-scale problems. It seems to me that NeuralPCG is much more expensive than existing SOTA algorithms. E.g. algebraic multi-grid.


12. The baseline preconditioning methods are very weak. In practice, we use much more efficient ones.

**Summary Of The Paper:**

This paper proposes NeuralPCG, which learns a preconditioner using neural networks to improve the solution procedure of
the linear solver for the resulting system of linear equations from discretizing partial differential equations (PDEs). The authors
tested the learned preconditioner for two- and three-dimensional simple PDEs.

**Summary Of The Review:**

The contribution of this paper is very limited. The paper ignores the entire existing research on learning preconditioners for solving systems of linear equations. Indeed, many existing works on deep learning of preconditioners are more solid in algorithm and validation than NeuralPCG. The experimental results are not convincing either.

---

> ### Author Response · Authors · 2022-11-16
> **Response to Reviewer G2sH**
>
> Dear Reviewer G2sH,
>
> We thank the reviewer for the valuable suggestions and comments. We address each of the reviewer’s concern here:
>
> - **Learning Baseline:** We thank the reviewer for bringing up these works \[1, 2] to our attention, and please refer to our general response about the comparison with these existing works.
>
> - **Novelty**: We thank the reviewers for their effort and time, please refer to the general response about the novelty of our method.
>
> - **AMG Baseline:** Thanks for bringing this up, indeed we are aware that AMG is a compelling method for solving Ax=b problems. Although in theory AMG is agnostic of the underlying geometric representation, in practice, people observed that AMG tends to work well on grid-like structures while struggling with unstructured geometric data \[3]. Previous work noticed (Algebraic) multigrid solvers may not be as effective as some standard linear numerical solvers, for example, direct solvers in elasticity problems discretized with tet or triangle meshes \[4] or MIC-CG iterative solver Poisson equations in fluid simulation \[5]. It turns out that making AMG a compelling solver on unstructured geometric representations is a non-trivial effort as Liu et al. demonstrates\[3]. For this reason, we consider our effort as a valuable contribution to the research community.
>
> - **Specialized numerical solver baseline:** There are highly specialized classical numerical methods for certain problems, we revised our paper by citing and discussing the differences.
>
>      We want to note that in our work we want to focus on the topic of preconditioners and iterative solvers. Specifically, how to learn a preconditioner that helps speed up iterative solvers. Therefore, we prioritize comparisons with other existing preconditioners in our paper.
>
> - **Convergence guarantee:** Our learned preconditioner indeed has convergence guarantee, which is also what sets us apart from other learning preconditioner works. Conjugate gradient algorithm is guaranteed to converge for second-order linear PDEs, as long as the devised preconditioner is a SPD matrix. Because our learned preconditioners are _SPD matrices by construction_, our proposed NeuralPCG are also guaranteed to converge.
>
> - **Condition number:** Here we show the condition number comparison as the reviewer requested. We note that our method reduces the condition number by a lot.
>
>  |                    |           |            |           |            |
>  | ------------------ | --------- | ---------- | --------- | ---------- |
>  |                    | Wave-2D   | Poisson-2D | Heat-2D   | Poisson-3D |
>   | No preconditioner  | 540272.25 | 43658.156  | 181.56293 | 1008.79425 |
>   | With our preconditioner  | 23.890734 | 8384.314   | 64.22761  | 136.86165  |
>  |                    |           |            |           |            |
>
> - **Universality:** Indeed, numerical methods can be applied to universal problems, but neural networks are still a valid contribution to the community. Learning based methods trades off bias and variance. E.g. If we have a problem on the same domain where we have to solve it multiple times, using a learning based method extracts bias for the given problem and grants a faster way to the solutions.
>
> - **Size of mesh and computational cost:** Indeed, mesh size is a critical factor for computational cost. However, we want to note that this increase in computational cost is universal to all methods, for both learning based methods and traditional numerical methods. The traditional sequential left-looking preconditioners such as IC are more prone to computation time increase with larger meshes, whereas learning-based methods are highly parallelizable, and thus suffer less from computation time increase. We do want to note that our method is limited by GPU memory, and cannot scale to arbitrarily large systems given limited computational resources.
>
> - **X-dependent PDE:** We thank the reviewer for mentioning this point. From what we understand, linear PDE cannot be x-dependent, if x is the unknown variable we are trying to solve, Non-linear PDEs indeed can be x-dependent, and we consider these x-dependent nonlinear pdes as future work.  Here we want to clarify that: a linear PDE is called linear if it is linear in the unknowns and its derivatives.
>
> - **Density of matrix L:** It is true that our proposed method preconditioners that follow the lower-triangular sparsity of A. We note that in practice, this is not a strong constraint. The state of the art simulation paper \[6] uses the Incomplete cholesky preconditioner, which follows the same structure.
>
> - **Training Time:** Training time is shown in section A.3 in Appendix Section. Here we want to note that the training convergence for our method is orders of magnitude faster compared to previous learning baselines, such as the suggested one \[1].

---

> > ### Author Response · Authors · 2022-11-16
> > **Continued Response to Reviewer G2sH**
> >
> > - **Generalize to different PDEs and different scales of Meshes:** In experiment section 4.5, we show that our method is generalizable to meshes of different scales and of different geometry. Existing learning-based preconditioners, like the ones that the reviewer suggested [1, 2], focus on specific problem, whereas we propose a more generic approach for all second-order linear PDEs, or SPD system matrix A. We do note that our method exploits data distribution of field values, and thus we focus on experiments that train and test on the under the same PDE setting. This strategy is commonly employed by existing works on learning-based preconditioners [1, 2] or state-of-the-art learning-based physics simulation [8, 9]. These prior works also train and test on the same PDE setting.
> >
> > - **1e-12 error threshold**: Different problems requires different accuracy threshold. Previous works [7] use $1e-16$ for their problem. We show various threshold levels from $1e-2 \sim 1e-12$ to provide a detailed comparison.
> >
> > - **Graph Oversmoothing:**  Oversmoothing typically happens with very deep graph neural networks. We do not observe the oversmoothing issue in our experiments, because we use only five layers of message passing on problem domains of 4,000 to 23,300 nodes.
> >
> > - **Reproducibility:** We will release both code and data in the near future.
> >
> > ****
> >
> > \[1] Johannes Sappl, Laurent Seiler, Matthias Harders, and Wolfgang Rauch. Deep learning of preconditioners for conjugate gradient solvers in urban water related problems, 2019.
> >
> > \[2] Jan Ackmann, Peter D. Duben, Tim N. Palmer, and Piotr K. Smolarkiewicz. Machine-learned preconditioners for linear solvers in geophysical fluid flows, 2020.
> >
> > \[3] Li, Minchen and Ferguson, Zachary and Schneider, Teseo and Langlois, Timothy R and Zorin, Denis and Panozzo, Daniele and Jiang, Chenfanfu and Kaufman, Danny M. Incremental potential contact: intersection-and inversion-free, large-deformation dynamics. ACM Trans. Graph. 2020.
> >
> > \[4] Robert Bridson. Fluid simulation for computer graphics.AK Peters/CRC Press. 2015
> >
> > \[5] Zangyueyang Xian, Xin Tong, Tiantian Liu. A Scalable Galerkin Multigrid Method for Real-time Simulation of Deformable Objects. ACM Transactions on Graphics 38(6) \[Proceedings of SIGGRAPH Asia],2019.
> >
> > \[6] Tao Du, Kui Wu, Pingchuan Ma, Sebastien Wah, Andrew Spielberg, Daniela Rus, Wojciech Matusik. ACM Transactions on Graphics 2022 (SIGGRAPH 2022). 2022
> >
> > \[7] Bolun Wang and Zachary Ferguson and Xin Jiang and Marco Attene and Daniele Panozzo and Teseo Schneider.  Fast and Exact Root Parity for Continuous Collision Detection, Computer Graphics Forum (Proceedings of Eurographics). 2022
> >
> > \[8] Tobias Pfaff, Meire Fortunato, Alvaro Sanchez-Gonzalez, Peter W. Battaglia. Learning Mesh-Based Simulation with Graph Networks. International Conference on Learning Representations (ICLR), 2021.
> >
> > \[9] Qingqing Zhao and David B. Lindell and Gordon Wetzstein, Learning to Solve {PDE}-constrained Inverse Problems with Graph Networks. ICML. 2022.

---

### Official Review · Reviewer_94zp · 2022-10-24

**Confidence:** 3
**Correctness:** 3
**Technical Novelty And Significance:** 3
**Empirical Novelty And Significance:** 3
**Recommendation:** 5

**Clarity, Quality, Novelty And Reproducibility:**

This paper is well-written and the overall quality is good.

The investigation on deep learning based preconditioner of PDE solver can date back to many years ago, some more similar paper can be found, e.g. "Machine Learning-Aided Numerical Linear Algebra: Convolutional Neural Networks for the Efficient Preconditioner Generation" and “Deep Learning of Preconditioners for Conjugate Gradient Solvers in Urban Water Related Problems”. This paper does not clarify differences from other similar papers, except the type of neural network used.

**Strength And Weaknesses:**

Strength:
The NeuralPCG combine the advantage of classic iterative solver and the machine learning based solver. It can have a good accurate solutions compared with neural network based PDE solver and work faster than classic solver.

Weaknesses:
The approach is now limited to linear PDEs with symmetric-positive-definite matrix, and the preconditioner’s sparsity is constrained to be the lower triangular sparsity of A.

**Summary Of The Paper:**

This paper proposes a so-called Neural-Network-Preconditioned Conjugate Gradient (NeuralPCG) method to solve partial differential equations. This method uses graph neural network to model the preconditioner of iterative solvers for PDE. It is noted that the NeuralPCG can solve PDE with a satisfying convergence condition and works faster than classic PDE solver. Several classic examples, including passion equation and heat conducted equation, are solved via the NeuralPCG, the comparing results with classic numerical PCG method show the generality and efficiency of the proposed method.

**Summary Of The Review:**

This is a well-written and easy to read paper. However, it did not review enough on the investigations of deep learning based precondition method. Some comments are given as follows:

Comment 1. Maybe the contour plot of Fig.1 and Fig.3 can have a bar of labeled field.

Comment 2: This paper shows the performance of NeuralPCG method based on the Conjungate Gradient method, can this paper give some discussions on the performance of general preconditioning method based on deep learning method?

Comment3: This paper only shows a little reviews on the investigations of deep learning based preconditioner method and does not clarify the distinguished difference of this paper with other similar paper.

Comment4: The accuracy of classic finite element method often shows dependence of the numbers of mesh, does this method also show this dependency?

---

> ### Author Response · Authors · 2022-11-16
> **Response to Reviewer 94zp**
>
> Dear Reviewer 94zp,
>
> We thank the reviewer for the valuable suggestions and comments. We address each of the reviewer’s concern here:
>
> - **Linear PDE:** In our work, we focus on the solving systems of Ax=b. In theory, if any problems can be written in the form of Ax=b, and A is SPD, they can benefit from our method. We are not specifically tied to linear pdes. For example, we show that our method can directly apply to the incompressible Navier-Stokes equation (under the standard operator splitting scheme) in the challenging cylindrical flow case (see poisson-2d), capturing the signature steady flow bifurcating into the vortex shedding regime.
>
> - **Density of matrix L:** It is true that our proposed method preconditioners that follow the lower-triangular sparsity of A. We note that in practice, this is not a strong constraint. The state of the art simulation paper \[2] uses the Incomplete cholesky preconditioner, which follows the same structure.
>
> - **Accuracy:** We thank the reviewer for this question. Here we want to clarify this point that might be confusing to many. The two accuracies, accuracy of discretization and accuracy of solver solutions, mean two different things:
>
>   - **1)** Accuracy associated with number of nodes in mesh or finite elements discretization accuracy means how the discretized system, such as mesh or grid, approximates the continuous problem. This accuracy is associated with the size of the $Ax=b$ system.
>   - **2)** Our method and all numerical or learning based methods deal with the accuracy of the solution, given the $Ax=b$ system. In mathematical terms, $x_{gt} = A^{-1}b$. And for any solutions obtained by learning-based methods or traditional iterative solvers, $x_{solve}$, this accuracy is $\frac{|x_{gt} - x_{solve} | }{ norm(x_{gt})}$
>
> - **Learning Baseline:** We thank the reviewer for bringing \[1] to our attention. Please refer to the general response about comparison with learning-baseline methods.
>
>
> \[1] Johannes Sappl, Laurent Seiler, Matthias Harders, and Wolfgang Rauch. Deep learning of preconditioners for conjugate gradient solvers in urban water related problems, 2019.
>
> \[2] Tao Du, Kui Wu, Pingchuan Ma, Sebastien Wah, Andrew Spielberg, Daniela Rus, Wojciech Matusik. ACM Transactions on Graphics 2022 (SIGGRAPH 2022). 2022

---

### Official Review · Reviewer_DNx8 · 2022-11-04

**Confidence:** 1
**Correctness:** 3
**Technical Novelty And Significance:** 3
**Empirical Novelty And Significance:** 3
**Recommendation:** 5

**Clarity, Quality, Novelty And Reproducibility:**

The paper is well written and easy to follow with clear comparison to the literature.

**Strength And Weaknesses:**

Strength:  The authors propose a novel and efficient NeuralPCG  solver for linear second-order PDEs and present a framework for studying the performance of preconditioners in classic iterative solvers.

Weaknesses: The paper is well written and easy to follow with clear comparison to the literature.  The significance and novelty of this paper are limited.

**Summary Of The Paper:**

The authors presented NeuralPCG, a hybrid ML-numeric PDE solver. While prior ML PDE solvers
often do not have precision guarantees, their approach attains arbitrary precision (up to floating-point
error) by construction.

**Summary Of The Review:**

The significance and novelty of this paper are limited.

---

> ### Author Response · Authors · 2022-11-16
> **Response to Reviewer DNx8**
>
> Dear Reviewer DNx8:
>
> We thank the reviewer for their effort and time. Please refer to the general response about the novelty of our method. We revised our paper to highlight the discussion about our novelty.

---

### Author Response · Authors · 2022-11-16
**General Response**

We thank the reviewers for the efforts and comments, we would like to highlight some shared concerns.

1. **novelty of our method over previous learning-based preconditioners:**

   1. We design a novel loss function that exploits the data distribution on field values that are neglected by prior approaches, e.g. [1]. Our loss function turns out to speed up by a large margin as reflected in section 4.4 and Table 2.
   2. We focus on mesh-based problem domains using GNN. To the best of our knowledge, prior learned based preconditioner methods all focus on grid and voxel data using CNN. We leverage the duality between graph and sparse matrices to ensure the positive-definiteness of the learned preconditioner, which guarantees convergence.
   3. Unlike previous learning based preconditioner works [1] that aim to solve a specific problem, e.g. Sappl et al. [1] for the Urban water problem, Akmann et al [2]. for geophysical fluid flows, we propose a more generic method that works on various second-order linear PDEs.

2. **comparison with learning-based baseline methods:**
      We thank the reviewer for bringing up these works [1] to our attention, and we compare our approach with this paper. We have revised our paper, adding the mentioned paper in the related work, and experiments in appendix. Here, we want to emphasize major differences between our method and the suggested baseline:

   1. The suggested learning baseline uses convolution neural networks on grid structured data, while we work with triangle / tetrahedral meshes with a graph neural network. Grid for structured geometry, and mesh is for irregular geometry boundary. For example, these baselines cannot be directly applied to any of the training data presented in this work. For this reason, we adopt their method (loss function and training scheme) but retain a GNN to work on meshes. We show the comparison here (more detailed comparison can be found in the revised manuscript):

        |Task           | Method         | precompute time (s)  | time (iter) 1e-10  |
        |---------------|-------------------|----------------------------|------------------------|
        |heat-2d      | NN baseline  | 0.0237                      |3.223 (200)         |
        |heat-2d      | ours               | 0.0237                     | **2.469 (152)**         |
        |wave-2d     | NN baseline | 0.0163                      | 0.165 (12)           |
        |wave-2d     | ours              | 0.0147                      | **0.156 (12)**            |
        |poisson-2d  | NN baseline | 0.0129                      |  2.221 (662)         |
        |poisson-2d  | ours             | 0.0145                      | **1.312 (374)**         |
        |                       |                    |                                    |                                 |
   2. We can see that NN baseline [1] works on par with our method on the wave-2d setting, but it does not work in many other settings, heat-2d, poisson-2d. For Poisson-2d setting the resulting convergence iteration is even worse than the pcg-jacobi method. Here, we want to note that the existing work is not originally designed to work on any second-order linear PDEs.
   3. Using condition number as the loss energy is not computationally efficient. Condition number by definition requires full eigen decomposition, which is $O(n^3)$. Computation scales cubically with problem size. We train the baseline methods for 3 days, 72 hours, and the method does not converge in poisson-3d setting, which uses a mesh with 23,300 nodes.


[1] Johannes Sappl, Laurent Seiler, Matthias Harders, and Wolfgang Rauch. Deep learning of preconditioners for conjugate gradient solvers in urban water related problems, 2019.

[2] Jan Ackmann, Peter D. Duben, Tim N. Palmer, and Piotr K. Smolarkiewicz.
Machine-learned preconditioners for linear solvers in geophysical fluid flows, 2020.

---

### Decision · Program_Chairs · 2023-01-20

**Decision:**

Reject

**Justification For Why Not Higher Score:**

The paper is weak on the comparison with state-of-the art sparse LU preconditioners. IC(0) is a very weak baseline.

**Justification For Why Not Lower Score:**

N/A

**Metareview: Summary, Strengths And Weaknesses:**

The paper proposes a neural architecture for learning preconditioners of the form $LL^{\top}$ for sparse positive definite matrices and $L$ is sparse. They also put the right hand side into the neural network design, as in the MeshGraphNet. The main difference is that they generate the matrix $L$ rather than the solution itself.

The idea of the paper is simple and easy to follow, and potentially this is a useful approach.

However, a lot of weaknesses have been pointed out by the reviewers, which I can confirm based on my own reading. First of all, most of the work on sparse Cholesky factorization seems to be unknown to the authors. Incomplete Cholesky factorization (also known as IC(0)) is a classical but not the most efficient method, see for example, second-order LU method by Igor Kaporin from 1998 [1]
Sparsekit software (Y. Saad) has the implementation of so-called ILU(2) preconditioner, which is based on the higher-order sparsity structure [2]. It might be a good idea to extend the idea in the paper for different sparsity pattern.
Finally, the loss function (approximation of the matrix on the right-hand sides) is not the optimal one. The preconditioner should minimize the condition number of the matrix. I suggest to utilize the approach in the paper [3] as the loss function.

[1] Kaporin IE. High quality preconditioning of a general symmetric positive definite matrix based on its UTU+ UTR+ RTU‐decomposition. Numerical linear algebra with applications. 1998 Nov;5(6):483-509.
[2] Chow E, Saad Y. Experimental study of ILU preconditioners for indefinite matrices. Journal of computational and applied mathematics. 1997 Dec 10;86(2):387-414.
[3] Oseledets I, Fanaskov V. Direct optimization of BPX preconditioners. Journal of Computational and Applied Mathematics. 2022 Mar 1;402:113811.